# ACCELERATED NEURAL NETWORK TRAINING WITH ROOTED LOGISTIC OBJECTIVES

## ABSTRACT

Many neural networks deployed in the real world scenarios are trained using cross entropy based loss functions. From the optimization perspective, it is known that the behavior of first order methods such as gradient descent crucially depend on the separability of datasets. In fact, even in the most simplest case of binary classification, the rate of convergence depends on two factors: 1. condition number of data matrix, and 2. separability of the dataset. With no further pre-processing techniques such as over-parametrization, data augmentation etc., separability is an intrinsic quantity of the data distribution under consideration. We focus on the landscape design of the logistic function and derive a novel sequence of *strictly* convex functions that are at least as strict as logistic loss. The minimizers of these functions coincide with those of the minimum norm solution wherever possible. The strict convexity of the derived function can be extended to finetune state-of-the-art models and applications. In empirical experimental analysis, we apply our proposed rooted logistic objective to multiple deep models, e.g., fully-connected neural networks and transformers, on various of classification benchmarks. Our results illustrate that training with rooted loss function is converged faster and gains performance improvements. Furthermore, we illustrate applications of our novel rooted loss function in generative modeling based downstream applications, such as finetuning StyleGAN model with the rooted loss. The code implementing our losses and models can be found here for open source software development purposes: https://anonymous.4open.science/r/rooted_loss.

## 1 INTRODUCTION

Neural networks have become a necessity to enable various real-world applications, especially in large scale settings. An appropriate parameterized model is chosen with the information of the domains or use-cases pertaining to the applications (Devlin et al. (2018); Radford et al. (2021); Caron et al. (2021)). Then, the parameters are iteratively modified to optimize a mathematically valid loss function applied on data points which represent the application under consideration (Goodfellow et al. (2016); Ghosh et al. (2017); Lin et al. (2017); Kavalerov et al. (2021); Hui et al. (2023)). Once the iterative procedure terminates (or is terminated with stopping conditions), the model parameters can be used to make predictions on unseen points. Thus, it is crucial to understand how different algorithms behave during optimization phase that correspond to the training procedure. In large scale setting, first order methods are preferred since they require the least computing resources, and are easier to implement with Automatic Differentiation packages (Paszke et al. (2017); Loshchilov & Hutter (2018); Reddi et al. (2019)). Naturally, the success and efficiency of first order methods depend on the landscape properties of the loss function when are applied on samples in datasets (Deng et al. (2009); Karras et al. (2019)).

**How does dataset affect optimization landscape?** Consider the task of classification in which a dataset $\mathcal{D}$ is represented as a set of pairs $(x, y)$, where $x$ denote features, and $y$ denote corresponding classes or labels (Pranckevičius & Marcinkevičius (2017); Singh et al. (2017); Zhang & Liu (2023)). In binary classification, the task is to categorize $x$ into one of two classes using model parameters after optimization. Here, it is known the rate of convergence of (stochastic) gradient descent – the de-facto first order method – to the optimal solution is primarily influenced by two factors: (1) *condition number* of the loss function (Overton (2001)): this number gives an insight into the structure and properties of the dataset. A lower condition number implies a better gradient directions

which makes optimization faster for first order methods (Hazimeh et al. (2022); Boob et al. (2023)). While using a one layer neural network, this condition number is determined by the so-called data matrix in which $(x, y)$ pairs are appropriately stacked as rows/columns; (2) recent work have shown that *separability* of $\mathcal{D}$ is an important factor to consider for modeling and training purposes (Shamir (2021); Tarzanagh et al. (2023)). Intuitively, separability is a measure of how easily a model can distinguish between two $x$'s from different classes $y$'s in $\mathcal{D}$. A highly separable dataset is easier to classify, and the optimization process is expected to converge faster. Indeed, separability is inherent to the dataset, and so without employing extra pre-processing steps like normalization (Ioffe & Szegedy (2015); Wu et al. (2021)), augmentation (Shorten & Khoshgoftaar (2019); Yarats et al. (2020)), over-parametrization (more than one layer) (Du et al. (2018); Buhai et al. (2020)), the level of separability is determined by the distribution from which $\mathcal{D}$ was sampled from.

Furthermore, the landscape of objective function that are used for generating or sampling points similar to $x$ have also been under investigation (Qi (2020)). A standard assumption in designing models or architectures for sampling is that $x$ is a smooth function – usually an image (or audio) considered as a two (or one) dimensional smooth function. With this assumption, various architectures have been proposed with (discrete) convolution or smoothing operators as the building blocks, such as DCGAN (Radford et al. (2015)), BigGAN (Brock et al. (2018)), StyleGAN (Karras et al. (2020b)). These smoothing based architectures called Generators gradually transform a random signal to $\tilde{x}$, a "fake" or synthetic sample. Then, a classification architecture called Discriminator is used to assign the probability of $\tilde{x}$ being a real sample from $\mathcal{D}$. While separability might not be the deciding factor in training the overall models, conditioning of loss functions used to train the Discriminator is crucial in determining the success of first order algorithms, and thereby the sampling process to obtain $\tilde{x} \sim x \in \mathcal{D}$ (Arora et al. (2017)).

**Our Contributions.** We provide a plug-in replacement for $\log$ based loss functions for supervised classification and unsupervised generation tasks with provable benefits. **First**, we show that there is a natural approximation to $(-\log)$ that is bounded from below that has the nice theoretical properties such as convexity and smoothness. Our novel result shows that the proposed *Rooted* loss with one additional parameter $k$ is at least as conditioned as $(-\log)$ based convex loss function, so provable acceleration. **Second,** we apply our loss to various datasets, architecture combinations and show that it can lead to significant empirical benefits for classifications. In image classifications, we show that the training time with our proposed rooted loss is much less than cross-entropy or focal loss. It also provides 1.44% - 2.32 % gains over cross-entropy loss and 5.78 % - 6.66 % gains over focal loss in term of test accuracy. **Third,** we apply rooted loss on generative models as downstream applications, showing lower FID and better generated images with limited training data.

## 2 PRELIMINARIES

Logistic regression is the task of finding a vector $w \in \mathbb{R}^d$ which approximately minimizes the empirical logistic loss (Ji & Telgarsky (2018)). While logistic regression can be seen as a single-layer neural network, deep neural networks contain multiple such layers stacked together. Each layer captures increasingly complex features from the input data. This hierarchical structure allows deep networks to model complex relationships.

Given datapoints $(x_i, y_i), i = 1, \ldots, n$, where $x_i \in \mathbb{R}^d$ denotes the features in $d-$ dimensions, and $y_i \in \{+1, -1\}$ is the binary label. By parametrizing the prediction function for a new sample $x$ as

$$f(x) := \mathbb{P}(y = \pm 1 | x) = \sigma(\pm w^\top x) \tag{1}$$

where $\sigma$ is the sigmoid function, the maximum likelihood estimator of $w \in \mathbb{R}^d$ can be obtained by minimizing the *negative* log-likelihood function of $w$ (Sur & Candès (2019)), written as,

$$\mathcal{L}_{\text{LR}}(w) := \frac{1}{n} \sum_{i=1}^{n} \log \left(1 + \exp \left(-y_i w^\top x_i\right)\right). \tag{2}$$

The cross-entropy (CE) loss is one of the most commonly used loss functions for training deep neural networks, most notably in multi-class classification problems. Given datapoints as $(x_i, y_{ik})$,

where $k \in c$, $c$ is the number of classes, $y_{ik} \in \{0, 1\}$ is a binary indicator of whether class $k$ is the correct classification for example $i$. Following the equation 2, multi-class cross-entropy loss is written as,

$$\mathcal{L}_{\text{CE}}(w) := -\frac{1}{n} \sum_{i=1}^{n} \sum_{k=1}^{c} y_{ik} \log \left( \frac{\exp(w_k^\top x_i)}{\sum_{j=1}^{c} \exp(w_j^\top x_i)} \right), \tag{3}$$

where $w_j^\top x_i$ represents the prediction score for the $i$-th example and the $j$-th class.

## 3 Rooted Logistic Objective Function

### 3.1 Motivation: From Logistic Objective to Rooted Logistic Objective

Logistic loss can serve as a smooth approximation to the element-wise maximum function, where smoothness is desirable in model design since gradient-based optimizers are commonly used. In this work, we consider to use the Taylor approximation of the natural logarithm function as follows: 1. for a fixed $u \in \mathbb{R}_+$, the derivative of $u^v$ is given by $u^v \log(u)$ by Chain rule, 2. now observe that by evaluating the derivative at $v = 0$, we obtain $\log(u)$, and 3. finally, plugging the above two in the definition of derivative we have that $\log(u) = \lim_{v \downarrow 0} \frac{u^v - 1}{v} = \lim_{k \uparrow \infty} k \left( u^{1/k} - 1 \right)$. Thus, for training purposes, we propose using a fixed sufficiently large $k$ with the following approximation to the log function: $\log(u) \approx k(u)^{\frac{1}{k}} - k$.

Here, the approximation seeks to express $\log(u)$ in terms of a function raised to the power of $\frac{1}{k}$. The constant $k$ provides a degree of freedom that can be adjusted to fine-tune the approximation. Building on this approximation, a novel loss function, termed the **R**ooted **L**ogistic **O**bjective function (**RLO**), is introduced. The key idea is to modify the traditional logistic loss by incorporating the above approximation. The loss function for this RLO can be defined as:

$$\mathcal{L}_{\text{RLO}}^k(w) = \frac{1}{n} \sum_{i=1}^{n} k \cdot \left[ l_i^k(w) := \left( 1 + \exp\left( -y_i w^\top x_i \right) \right)^{\frac{1}{k}} \right]. \tag{4}$$

**Intuition to prefer Rooted Loss over Log based losses.** Logistic loss plays a pivotal role in penalizing prediction errors, particularly for the true class denoted as $y_i$ in classification tasks. One of its notable characteristics is the high loss and large gradient when the function $f(x)$ approaches zero. This sharp gradient is beneficial in gradient-based optimization methods, such as gradient descent, because it promotes more significant and effective update steps, leading the convergence towards the optimal solutions. Moreover, when we consider the gradient contributions from incorrect classes, the "signal" coming from the gradient is weaker, so such optimization schemes may be less effective in driving the probabilities for these classes to zero. Specifically, optimization algorithms might struggle or take longer to drive the predicted probabilities of these incorrect classes towards zero. In simpler terms, while the logistic loss is adept at penalizing mistakes for the true class, it might be gentler or slower in correcting overconfident incorrect predictions. The deep neural networks (DNNs) trained by the softmax cross-entropy (SCE) loss have achieved state-of-the-art performance on various tasks (Goodfellow et al. (2016)).

### 3.2 Convexity of RLO

Standard logistic regression function in equation 2 has favorable convexity properties for optimization. In particular, it is strictly convex with respect to parameters $w$, for more details, see (Freund et al. (2018)). By direct calculation of Gradient and Hessian using Chain and Product rules, we obtain the gradient $\nabla_w l_i^k$ for a single point $(x_i, y_i)$,

$$\nabla_w l_i^k(w) = \frac{1}{k} [(1 + \exp\left( -y_i w^\top x_i \right))^{\left( \frac{1}{k} - 1 \right)} \cdot \exp\left( -y_i w^\top x_i \right))] \cdot (-y_i x_i) \tag{5}$$

$$= l_i^k(w) \cdot \frac{\exp\left( -y_i w^\top x_i \right)}{1 + \exp\left( -y_i w^\top x_i \right)} \cdot (-y_i x_i) = l_i^k(w) \cdot \frac{1}{\exp\left( -y_i w^\top x_i \right) + 1} \cdot (-y_i x_i) \tag{6}$$

$$= -g(w, x_i) \cdot y_i x_i, \tag{7}$$

where $g(w, x_i) := \sigma(y_i w^\top x_i) \cdot l_i^k(w) \geq 0$. Similarly, we obtain the second-order gradient $\nabla^2 l_i^k(w)$ for a single point $(x_i, y_i)$ as follows,

$$\nabla^2 l_i^k(w) = h(w, x_i) \cdot x_i x_i^\top, \tag{8}$$

where $h(w, x_i) := l_i^k(w) \cdot \sigma(y_i w^\top x_i) \cdot \left[1 - \sigma(y_i w^\top x_i) \cdot (1 - 1/k)\right] > 0$ since both $\sigma(\cdot), 1/k \in (0, 1)$. We have included the full proof of hessian in the Appendix A.1. With these calculations, we have the following result:

**Lemma 1** $\mathcal{L}_{RLO}^k(w)$ *is a strictly convex function whenever* $k > 1$ *as is considered here.*

Note that, our result is novel because standard composition rules for convex optimization do not apply. This is due to the fact the function $(\cdot)^{\frac{1}{k}}$ is a *concave* function in the nonnegative orthant. Numerically, the main advantage is that the condition number of $\mathcal{L}_{RLO}^k(w)$ is independent of data, while $\mathcal{L}_{RLO}^k(w)$ can be quite ill-conditioned for inseparable datasets due to the $\log(\cdot)$ function. More details can be found in Chapter 12 of (Overton (2001)).

While strict convexity is true for both Logistic and RLO loss functions, the following result says that the full batch RLO is guaranteed to be as conditioned as Logistic objective function by comparing the coefficient of the hessian term $x_i x_i^\top$ in RLO and Logisitic objectives (LO):

**Lemma 2** *Let* $r_i := h_{RLO}(w_i^*, x_i^*)/h_{LO}(w, x_i) \in \mathbb{R}_{\geq 0}$, *, where* $w_i^*$ *is the optimal parameters for sample* $i$. *Then if* $k \leq \exp(l_i^k(w_i^*))$ *then* $r_i > 1$.

Above, lemma 2 states that as long as $k$ is not chosen to be too large, the gradient directions may provide sufficient descent needed for fast convergence. This property makes it ideal for solving classification problems. From lemma 1 and 2, we can conclude that there is a range of values of $k$ that provides better conditioning for individual data points. It is beneficial when using stochastic algorithms that use a random mini-batch of samples at each iteration instead of the full dataset to compute gradient.

**Generalization properties of RLO.** Assuming that points $x_i \in \mathbb{R}^d$ are bounded i.e., $\|x\| \leq B_x$ and that there is an bounded optimal solution $\|w\| \leq B_o$, we expect that the generalization bounds for LR in equation 2 to hold for RLO in equation 4. This is because of the fact that asymptotically – when $k \uparrow +\infty$ – the hessian coefficient of RLO is at most 1, which guarantees that the gradient is lipschitz continuous (Lei et al. (2019); Bartlett & Mendelson (2002)).

### 3.3 Applying RLO for Generative Models

Generative models were studied as *statistical* problem where the goal is, given a training dataset $x_i, i = 1, 2...n$, learn a parametric model of its distribution $p(x)$. For an appropriate parametric model $f_\theta$, we need $\theta$ such that $f_\theta(z) \approx x$, where $z$ is usually a Gaussian vector to approximate some $x_i$ through the transformation $f_\theta$. For sampling, given a mapping $f_\theta$, synthetic data points can be generated by sampling a Gaussian vector $z$ and computing $f_\theta(z)$. This overcomes some of the architectural restrictions of $f_\theta$. This property is leveraged to come up with Generative Adversarial Networks (GANs), see Chapter 10 in (Lindholm et al. (2022)).

GANs are a class of models that help the synthesize data points from the model using $f_\theta$ which gets a Gaussian vector $z$ as an input. GANs are trained by comparing these synthetic samples with real samples from the training data $x_i$. The comparison is done by a critic, e.g., a binary classifier $g_\eta$ which judges the authenticity of the samples. It is an adversarial game where the generator's parameters $\theta$ are continuously updated to synthesize data close to reality while the classifier such as the discriminator wants to label them correctly as fake. The result is a generator that has successfully learned to generate data that the discriminator labels as real. The generator tries to maximize the classifier loss with respect to $\theta$ while the classifier tries to minimize the loss with respect to $\eta$. This leads to a rooted minmax problem with loss that is similar equation 4, written as,

$$\min_\theta \max_\eta V_k(f_\theta, g_\eta) = \mathbb{E}_{x \sim p_{data}(x)}[k \, (g_\eta(x))^{1/k}] + \mathbb{E}_{z \sim p_z(z)}[k \, (1 - g_\eta(f_\theta(z)))^{1/k}]. \tag{9}$$

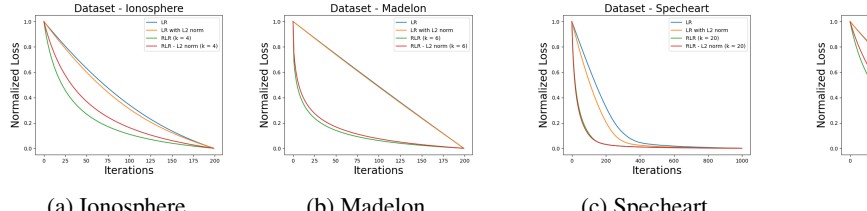

(a) Ionosphere      (b) Madelon      (c) Specheart      (d) Wine

Figure 1: The rate of convergence over iterations of standard logistic regression and RLO. The lines for the rooted logistic regression show the convergence for the value of k which gives the best test accuracy, $k = 4$ for Ionosphere, $k = 6$ for Madelon, $k = 20$ for Specheart and $k = 3$ for Wine. RLO converges faster than standard logistic regression in all the settings.

## 4 EXPERIMENTS

In this section, we illustrate the experiments of using our proposed RLO on multiple architectures of models on various benchmark datasets. Specifically, we compare rooted logistic regression with standard logistic regression on synthetic dataset and 4 benchmark datasets from UCI machine learning repository. Furthermore, we evaluate rooted loss against cross-entropy loss and focal loss by training state-of-the-arts deep models, e.g., ResNet (He et al. (2016)), ViT (Dosovitskiy et al. (2020)) and Swin (Liu et al. (2021)), on image classification tasks. Finally, we showcase the application of image generations using RLO with StyleGAN (Karras et al. (2020a)).

### 4.1 DATASETS

**Synthetic dataset Setup (Hui et al. (2023)):** We use a version of the popular 2-class spiral with 1500 samples, and we use 70% data for training and the remaining 30% data for testing.

**Dataset for regression:** The empirical studies are conducted on the following 4 benchmark classification datasets, which can be found in the publicly available UCI machine learning repository (Asuncion & Newman (2007)): Wine, Ionosphere, Madelon and Specheart.

**Image datasets:** We conduct image classification experiments to test the performance of rooted loss. In particular, we use CIFAR-10/100 (Krizhevsky et al. (2009)) for training from the scratch ,and Tiny-ImageNet (mnmoustafa (2017)) and Food-101 (Bossard et al. (2014))for finetuning. For our image generation experiments with StyleGAN, we use FFHQ dataset (Karras et al. (2018)) and the Stanford Dogs dataset (Khosla et al. (2011)). More data information are in Appendix A.2.

### 4.2 SHALLOW LOGISTIC REGRESSION VS ROOTED LOGISTIC REGRESSION

**Experiments setups:** The baseline is standard logistic regression. To showcase the benefits of RLO, we run the experiments with different numbers of k$\in [3, 20]$ for the proposed rooted logistic regression. Note that, for all the datasets except Specheart, we use the same number of iterations (200) and learning rate (0.01) for all the experiment settings. For Specheart, we increase the number of iterations to 1000 for better convergence and higher accuracy. We also evaluate standard logistic regression as well as RLO with/without $\ell_2$ regularization. More setup details are in Appendix A.4.2.

**Convergence analysis:** As mentioned above, we keep the experimental settings the same across all datasets, except Specheart. Figure 1 shows the convergence performance for Ionosphere, Madelon, Specheart and Wine datasets respectively. For all the datasets we can clearly see that RLO has better convergence performance compared to the standard logistic regression. We can see that the RLO, with and without $\ell_2$ regularization converge quicker than standard logisitc regression, and RLO without $\ell_2$ regularization converging comparatively faster. For the convergence results for other values of k, in the case of RLO, please refer Appendix A.4.2.

**Performance gains:** Table 1 shows the test accuracy for all the datasets under the different regression settings. For RLO, we also show the top 3 $k$ values which achieved the highest accuracy. As seen in the table, for all the datasets, RLO with/without $\ell_2$ regularization outperforms standard logistical regression with/without $\ell_2$ in term of accuracy on test set. Specifically, RLO with $\ell_2$ regu-

| | LR | LR - L2 | | RLO | RLO - L2 |
|---|---|---|---|---|---|
| Dataset | Test Acc. | Test Acc. | k | Test Acc. | Test Acc. |
| Wine | $90 \pm 4.15$ | $89.44 \pm 5.66$ | 3 | $\mathbf{97.22 \pm 1.75}$ | $94.55 \pm 3.51$ |
| | | | 11 | $82.77 \pm 1.23$ | $\mathbf{95.55 \pm 2.22}$ |
| | | | 13 | $91.66 \pm 5.55$ | $\mathbf{95 \pm 5.09}$ |
| Ionosphere | $81.4 \pm 2.73$ | $83.94 \pm 2.1$ | 4 | $85.07 \pm 1.12$ | $\mathbf{86.47 \pm 1.12}$ |
| | | | 3 | $\mathbf{86.47 \pm 1.69}$ | $85.63 \pm 0.56$ |
| | | | 16 | $84.5 \pm 0.00$ | $\mathbf{86.19 \pm 0.56}$ |
| Madelon | $52.03 \pm 1.9$ | $50.83 \pm 1.51$ | 6 | $\mathbf{54.36 \pm 0.71}$ | $52.75 \pm 0.97$ |
| | | | 9 | $\mathbf{54.13 \pm 0.58}$ | $51.8 \pm 1.43$ |
| | | | 19 | $52.36 \pm 1.14$ | $\mathbf{54.13 \pm 1.42}$ |
| Specheart | $80.49 \pm 3.92$ | $88.25 \pm 1.69$ | 20 | $84 \pm 3.10$ | $\mathbf{88.5 \pm 1.83}$ |
| | | | 15 | $82.75 \pm 1.83$ | $\mathbf{88 \pm 1.49}$ |
| | | | 13 | $82.99 \pm 2.44$ | $\mathbf{88 \pm 1.00}$ |

Table 1: Testing Accuracy from 5-Fold Cross Validation, using Shallow Logistic regression vs Rooted Logistic regression (RLO). Top 3 values of $k$ are shown for RLO. RLO with/without $\ell_2$ regularization outperforms Shallow Logistic regression with/without $\ell_2$, in term of accuracy on test sets of all 4 datasets.

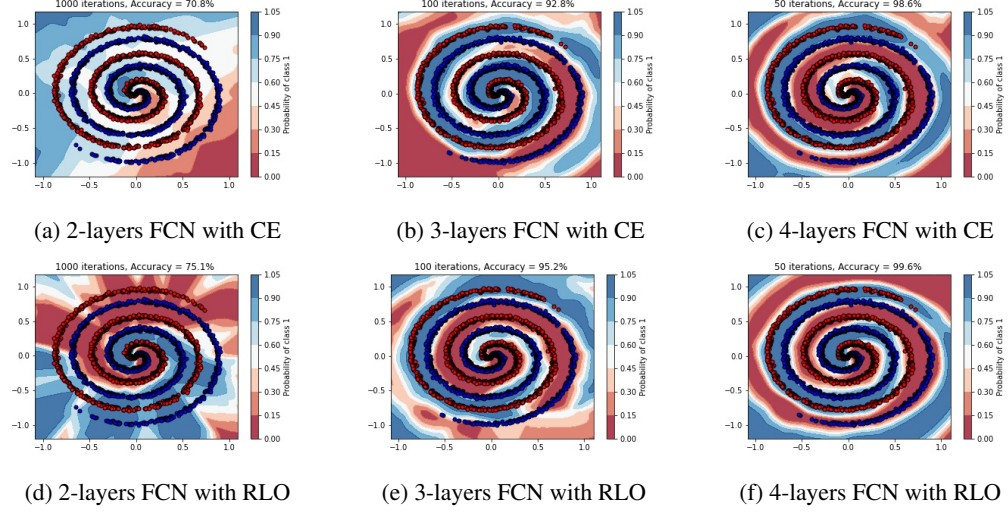

(a) 2-layers FCN with CE

(b) 3-layers FCN with CE

(c) 4-layers FCN with CE

(d) 2-layers FCN with RLO

(e) 3-layers FCN with RLO

(f) 4-layers FCN with RLO

Figure 2: The color demonstrated the estimated probability of a class label being identified as 1, as aligned with the scale located on the right side in the figures. The intervening white line between the red and blue regions denotes the decision boundary, In (a), (b) and (c), we train a 2-layer FCN for 1000 iterations, a 3-layer FCN for 100 iterations, and a 4-layer FCN for 50 iteration with cross-entropy loss. In (d), (e) and (f), we train a 2-layer FCN for 1000 iterations, a 3-layer FCN for 100 iterations, and a 4-layer FCN for 50 iteration with rooted logistic objective loss.

larization consistently achieves higher accuracy rates for different values of $k$. Hence, we conclude that our proposed RLO is beneficial to accelerate the training and also provide improvements.

## 4.3 DEEP NEURAL NETWORK FOR CLASSIFICATION WITH RLO

**Experiments setups:** At first, we implemented three different layers (2, 3, 4) fully-connected neural networks (FCN) on synthetic dataset. The training iterations are 1000, 100, and 50 respectively. We use the same hidden size of 100, learning rate as 0.01 and k of 3 for three FCNs. For the vision models in image classification tasks, as multi-class classification, we train and finetune on ViT-B (Dosovitskiy et al. (2020)), ResNet-50 (He et al. (2016)), and Swin-B (Liu et al. (2021)) models. The k parameters of our proposed RLO are chosen from the set $\{5, 8, 10\}$. We train on CIFAR-10 and CIFAR-100 for 200 epochs with ViT and 100 epochs with ResNet and Swin. Moreover, we finetune these models on Tine-ImageNet and Food-101 for 10 epochs. We train and fine-tune both

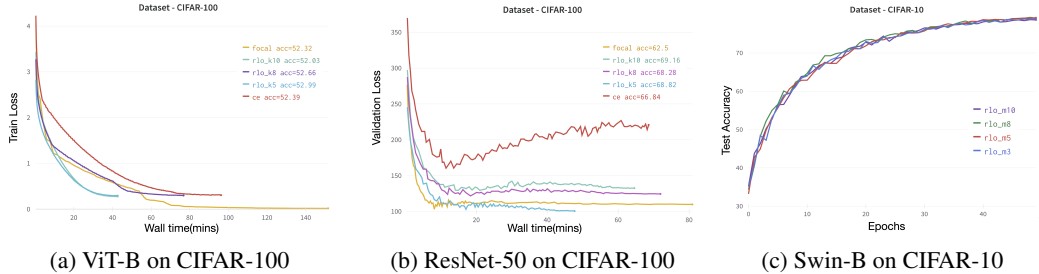

(a) ViT-B on CIFAR-100  (b) ResNet-50 on CIFAR-100  (c) Swin-B on CIFAR-10

Figure 3: RLO performance on CIFAR-100 training with different models. The x-axis is wall time in minutes. RLO obtains more stable validation loss, and use less time for training on all models.

| Dataset | Model | CE | | Focal | | RLO-5 | | RLO-8 | | RLO-10 | |
|---|---|---|---|---|---|---|---|---|---|---|---|
| | | Time | Acc | Time | Acc | Time | Acc | Time | Acc | Time | Acc |
| CIFAR-10 | ViT | 12.42 | 79.15 | 62.16 | 77.78 | 12.67 | 79.1 | 12.80 | **79.64** | 12.70 | 79.33 |
| | ResNet | 24.13 | 87.67 | 110.98 | 86.50 | 22.22 | 88.54 | 19.81 | 88.53 | 21.27 | **88.79** |
| | Swin | 20.95 | 80.99 | 22.49 | 80.01 | 20.96 | 81.52 | 20.75 | **81.91** | 20.95 | 80.9 |
| CIFAR-100 | ViT | 48.58 | 52.39 | 61.83 | 52.32 | 12.58 | **52.99** | 12.73 | 52.67 | 12.59 | 52.03 |
| | ResNet | 25.46 | 66.84 | 20.75 | 62.5 | 25.67 | 68.82 | 52.75 | 68.28 | 25.76 | **69.16** |
| | Swin | 20.14 | 53.6 | 63.24 | 53.66 | 20.64 | **53.91** | 20.03 | 53.29 | 20.27 | 53.54 |
| Tiny-INet | ViT | 920.94 | 84.7 | 821.65 | 85.08 | 901.68 | **86.05** | 908.69 | 85.73 | 905.26 | 85.38 |
| | ResNet | 253.92 | 73.39 | 245.74 | 73.95 | 257.85 | **74.19** | 259.85 | 74.05 | 255.94 | 74.1 |
| | Swin | 950.72 | 88.85 | 952.54 | 88.22 | 932.56 | 88.88 | 928.37 | 88.74 | 926.53 | **88.91** |
| Food-101 | ViT | 670.43 | 80.59 | 677.32 | 79.07 | 664.05 | 80.39 | 664.85 | 80.7 | 667.96 | **80.72** |
| | ResNet | 187.94 | 73.39 | 180.35 | 72.44 | 189.56 | 73.73 | 189.75 | **73.97** | 183.87 | 73.91 |
| | Swin | 718.01 | 87.21 | 700.89 | 86.64 | 723.89 | 87.53 | 723.25 | **87.64** | 717.04 | 87.52 |

Table 2: Test performance for image classifications on different datasets. Time is averaged one epoch training time in seconds. Note that, CE is cross-entropy for short. $k$ values are 5, 8, and 10. Our RLO obtains the best and second best accuracy in all datasets and models.

on 3 NVIDIA RTX 2080Ti GPUs. To evaluate our proposed RLO, we use cross-entropy (CE) loss and focal loss as baselines. More implementation details are in Appendix A.3.

**Observations on FCNs decision boundaries:** To enable interpretative understanding, we use the synthetic setups to visualize the decision boundaries learned by RLO compared with CE. Figure 3 shows the decision boundaries obtained from RLO and CE for three different FCNs training for different iterations. The intervening white line between the red and blue regions denotes the decision boundary, a critical threshold distinguishing classifications within the model. Specifically, comparing (b) and (e), and (c) and (f), we observe the margins which are the distances from datapoints to the decision boundary are larger for RLO in most of regions. Hence, RLO is beneficial to separate data points that enable the faster convergence rate.

**Nonconvex Optimization Benefits:** *(1) Performance gains:* we evaluate the effectiveness of RLO on nonconvex optimizations. In Figure 3, training with RLO for FCNs outperforms CE in term of accuracy in all different settings. It provides 1% - 4.3 % improvements for the binary classification. Furthermore, we illustrate the results of RLO on multiple image classification benchmarks. Table 2 shows that RLO performs the best and the second best accuracy across all datasets and network architectures. Specifically, training with RLO brings roungly 1.44% - 2.32 % gains over CE and 5.78 % - 6.66 % gains over focal in term of test accuracy. Additionally, Figure 3a and 3b show the training time of RLO are significantly less than CE and focal on different models. For example, the training wall time of ViT on CIFAR-100 for 200 epochs is 54 minutes and 109 minutes less than CE and focal respectively. Therefore, our proposed RLO can accelerate neural networks training and also provide performance improvements regardless of datasets and model architectures. *(2) Effects on overfitting:* Figure 3b shows the validation loss using CE is increasing over iterations. However, we observe that validation loss with RLO is decreasing over time on the same dataset and model, which is beneficial to reducing overfitting.

## 4.4  GAN-RELATED WITH RLO

**Experiments setups:** For the image generation setup, we use the version of StyleGAN capable of being trained by limited training data, as proposed by (Karras et al. (2020a)). All training is done on 3 NVIDIA RTX 2080Ti GPUs with FFHQ and Stanford Dogs dataset. We evaluate the effectiveness of RLO by replacing the original loss and compare it to StyleGAN's CE loss, for different values

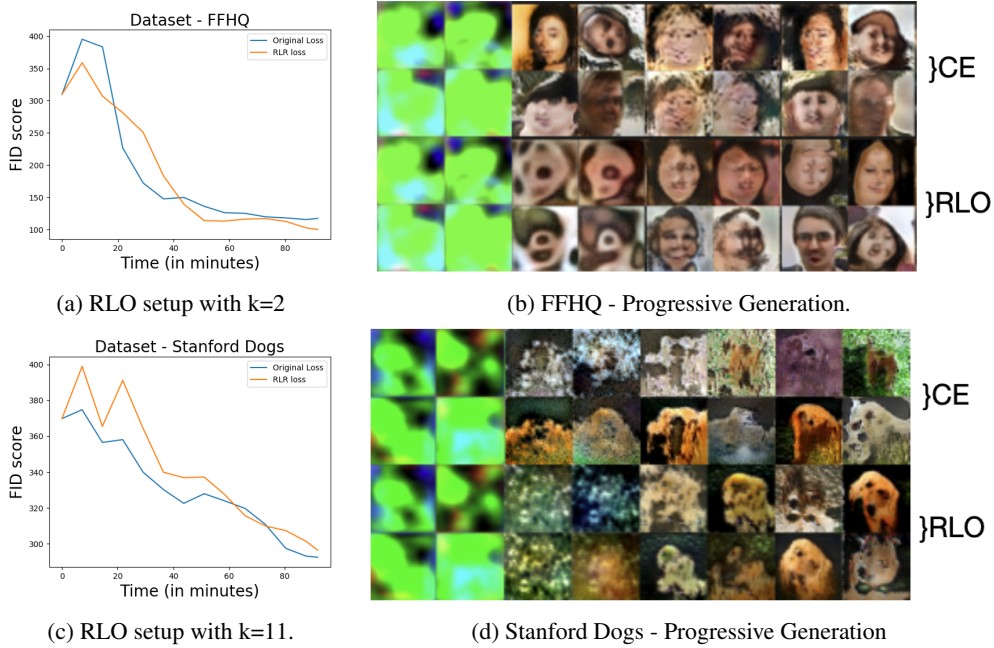

Figure 4: Results on FFHQ dataset and Stanford Dogs dataset. (a) FID score vs training time for both cross-entropy loss and RLO-2 setup. (c) FID score vs training time for both cross-entropy loss and RLO-11 setup. In (b) and (d), the top $8 \times 2$ row contains four instances of the image generation (Each image is a part of the $2 \times 2$ grid containing four images) using CE loss. The bottom $8 \times 2$ represents the same with the RLO setup at the same instances.

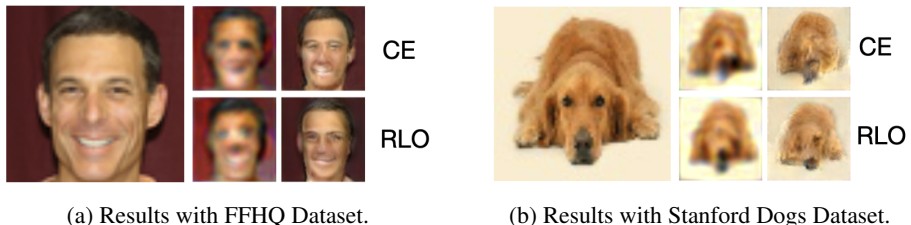

(a) Results with FFHQ Dataset.          (b) Results with Stanford Dogs Dataset.

Figure 5: For each setup, the target image can be seen on the left-most image. To its right, the first row shows the generated images with the projection obtained from the initial and final stages of the training respectively, with CE. The second row is the result of replacing it with RLO.

of $k$. To compare the efficacy of the models trained using RLO and CE loss, we take a large image from the original dataset, and compute its projection on the latent space using our model snapshots from the initial and final stages of the training. We then use these projections to generate an image using their respective models. More implementation details are in Appendix A.3.

**Observations:** As shown in Figure 4a, the setup with RLO trained on the FFHQ dataset produces a lower FID, which means better quality images, than the one with the CE. The progressive image generation while training these models are illustrated in Figure 4b. Images produced by RLO (bottom 2 rows) seems to be slightly better at the final stages of the training. Similar FID vs time comparison and progressive image generation for the Stanford Dogs dataset is shown in Figure 4, where the FID scores for the two models are close. Finally, in Figure 5, for a target image, we show the images obtained from the projections using the initial and final stages of training. For both FFHQ and Stanford Dogs dataset, we can see that the images generated using the final stage RLO models (bottom image, in the last column), produce details that are closer to the target image than CE. More generated images are shown in Appendix A.4.4.

| Model | # Param | CE | | Focal | | RLO-10 | |
|---|---|---|---|---|---|---|---|
| | | Train | Test | Train | Test | Train | Test |
| ResNet-34 | 21.3M | 99.76 | 89.92 | 94.04 | 85.79 | **99.79** | **89.98** |
| ResNet-50 | 23.7M | **99.47** | 86.65 | 93.98 | 80.13 | **99.47** | **86.72** |
| ResNet-101 | 42.7M | 99.69 | 84.28 | 94.40 | 77.75 | **99.74** | **85.12** |
| ViT-S | 14.4M | 67.17 | 67.7 | 66.51 | 67.74 | **68.31** | **68.37** |
| ViT-B | 85.1M | 72.49 | 72.12 | 71.61 | 71.45 | **73.23** | **72.41** |
| ViT-L | 226.8M | 76.56 | **74.81** | 74.56 | 73.72 | **77.17** | **74.81** |

Table 3: Ablations on model architectures, including running time and test performance on CIFAR-10. Time is averaged one epoch time in seconds. Note that, CE is cross-entropy for short. $k$ value is 10. Our RLO obtains the best train and test accuracy in all models.

### 4.5 ABLATION STUDIES

**More experiments on Parameter family:** Our proposed RLO have hyperparameter as $k$ shown in equation 4. We conduct experiments on training models with different values of $k$. As results shown in Table 1 and 2, and Figure 4a and 4c, the best $k$ values vary on datasets and neural network architectures. Moreover, we observe that $k$ is much more smaller than number of samples or feature dimensions as in lemma 2. In addition, we extend our RLO parameter family to $k$ and $m$, which $m$ is the multiplier in equation 4. Figure 3c shows the test accuracy over iterations using different values of $m$. Note that, we train Swin using different $m$ on CIFAR-10 with $k = 8$. The performances on different $m$ are similar, and $m = 8$ is slightly better than others in term of test accuracy. More results of different $k$ values are in Appendix A.4.

**Is RLO sensitive to model architectures/sizes?** First, in Table 2, we showed the performance of RLO for image classification tasks for various combinations of datasets and deep neural network models. We saw that for k values from the set $\{5, 8, 10\}$ achieved performance gain over CE and focal methods, in terms of test accuracy, for almost all the settings. For further ablation, we chose the $k$ value to be 10 and compare with the baselines under different model architectures of ResNet and ViT. The ablation results in Table 3 suggest that RLO resoundingly performs better, even under different architectures of the same model. Both the training as well as the test accuracy under RLO-10 are better than those trained with CE and focal losses. Thus suggesting that RLO guarantees performance gain across different hyperparameters and model architectures.

## 5 CONCLUSIONS AND FUTURE WORK

We presented comprehensive evaluations of a new class of loss functions for prediction problems in shallow and deep network. This class of loss functions has many favorable properties in terms of optimization and generalization of learning problems defined with high-dimensional data. Recent results suggest that standard logistic loss $L_{\text{LR}}(\cdot)$ need to be adjusted for better convergence and generalization properties (Sur & Candès (2019)). By taking limit as $k \uparrow +\infty$, or equivalently $1/k \downarrow 0$ (say using L'Hôpital's rule), we can see that $\lim_{k \to \infty} L_{\text{RLO}}^k(\cdot) = L_{\text{LO}}(\cdot)$. Moreover since $L_{\text{RLO}}^k(\cdot)$ and first order necessary condition are both smooth with respect to $k$, the minimizers also coincide in the limit. We leave rate aspects of convergence to max classifier and generalization aspecst of obtained solution as in (Soudry et al. (2018); Freund et al. (2018)) for future work. Finally, dependence of the parameter $k$ on excess risk and generalization bounds for RLO is also left as a future work. We believe insights from recent generalized linear models are fruitful directions to pursue (Hanneke et al. (2023); Emami et al. (2020)).

Our investigations show that the rooted logistic loss function performs better when using first-order methods. However, the convergence guarantees for first-order methods are relatively weak for pre-training architectures with a large number of parameters, such as vision models. Moreover, since these models have sequential aspects in their training formulations, the convergence rate is further reduced in practice. Therefore, it would be interesting to consider second-order methods like Sophia Liu et al. (2023), to optimize $\mathcal{L}_{\text{RLO}}^k$ for some $k$.

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
