## A APPENDIX

### A.1 PROOF OF LEMMA 2.

We first simplify gradient and hessian of LR in equation 2 and RLO in equation 4 one by one.

Recall that the LR of a single sample $(x_i, y_i) \in \mathbb{R}^d \times \{+1, -1\}$ is given by,

$$\mathcal{L}_{\text{LR}}(w; x_i, y_i) = \log\left(1 + \exp\left(-y_i w^\top x_i\right)\right). \tag{10}$$

By using chain rule to differentiate loss in equation 10 with respect to $w$, we obtain the gradient $\nabla_w \mathcal{L}_{\text{LR}}(w; x_i, y_i)$ as,

$$\nabla_w \mathcal{L}_{\text{LR}}(w; x_i, y_i) = -y \left[\frac{\exp(-y_i w^\top x_i)}{1 + \exp(-y_i w^\top x_i)}\right] x_i = -y_i \left[\frac{1 + \exp(-y_i w^\top x_i) - 1}{1 + \exp(-y_i w^\top x_i)}\right] x_i$$

$$= -y_i \left[1 - \frac{1}{1 + \exp(-y_i w^\top x_i)}\right] x_i. \tag{11}$$

Now applying the quotient rule and then chain rule once again to each coordinate of the gradient function in equation 11 and arranging the columns appropriately we obtain the hessian $\nabla^2 \mathcal{L}_{\text{LR}}(w; x_i, y_i)$ as,

$$\nabla^2 \mathcal{L}_{\text{LR}}(w; x_i, y_i) = -y_i \left[-y_i \cdot \frac{\exp(-y_i w^\top x_i)}{\left(1 + \exp(-y_i w^\top x_i)\right)^2}\right] x_i x_i^\top$$

$$= \left[\frac{\exp(-y_i w^\top x_i)}{\left(1 + \exp(-y_i w^\top x_i)\right)^2}\right] x_i x_i^\top \quad \left(\text{since } y_i^2 = 1\right). \tag{12}$$

Moreover, since $z^\top x_i x_i^\top z = \left(z^\top x_i\right)^2 \geq 0$ for any $z \in \mathbb{R}^d$, and $\exp(\cdot) > 0$ for any finite argument, we have that the hessian $\nabla^2 \mathcal{L}_{\text{LR}}(w; x_i, y_i$ is positive definite, and so $\mathcal{L}_{\text{LR}}$ is a strictly convex function.

We now repeat the calculations for RLO in similar way. Recall that the RLO of a single sample single sample $(x_i, y_i) \in \mathbb{R}^d \times \{+1, -1\}$ is given by,

$$\mathcal{L}_{\text{RLO}}^k(w; x_i, y_i) = k \cdot \left(1 + \exp\left(-y_i w^\top x_i\right)\right)^{\frac{1}{k}}. \tag{13}$$

By using chain rule to differentiate loss in equation 13 with respect to $w$, we obtain the gradient $\nabla_w \mathcal{L}_{\text{RLO}}^k(w; x_i, y_i)(w; x_i, y_i)$ as,

$$\nabla_w \mathcal{L}_{\text{RLO}}^k(w; x_i, y_i) = -y \left[\frac{\exp(-y_i w^\top x_i)}{\left(1 + \exp(-y_i w^\top x_i)\right)^{1 - \frac{1}{k}}}\right] x_i = -y_i \left[\frac{1 + \exp(-y_i w^\top x_i) - 1}{\left(1 + \exp(-y_i w^\top x_i)\right)^{1 - \frac{1}{k}}}\right] x_i$$

$$= -y_i \left[\left(1 + \exp(-y_i w^\top x_i)\right)^{\frac{1}{k}} - \left(1 + \exp(-y_i w^\top x_i)\right)^{\frac{1}{k} - 1}\right] x_i. \tag{14}$$

Now applying the quotient rule and then chain rule once again to each coordinate of the gradient function in the two terms in equation 14, arranging the columns appropriately and simplifying, we obtain the hessian $\nabla^2 \mathcal{L}_{\text{RLO}}(w; x_i, y_i)$ as,

$$\nabla^2 \mathcal{L}_{\text{RLO}}(w; x_i, y_i) = \frac{\exp(-y_i w^\top x_i)}{\left(1 + \exp\left(-y_i w^\top x_i\right)\right)^{1 - \frac{1}{k}}} \cdot \left[\frac{1}{k} + \frac{\left(1 - \frac{1}{k}\right)}{1 + \exp(-y_i w^\top x_i)}\right] \cdot x_i x_i^\top. \tag{15}$$

Once again, all the coefficents are positive since $1/k < 1$, and so $\nabla^2 \mathcal{L}_{\text{RLO}}(w; x_i, y_i)$ is positive definite. Whence, $\mathcal{L}_{\text{RLO}}^k(w; x_i, y_i)$ is a strictly convex function for any $k > 1$.

We are now ready to compare the scalar coefficients of hessian of LR in equation 12 and RLO in equation 15. As the first step, we note that,

$$\frac{1}{k} + \frac{\left(1 - \frac{1}{k}\right)}{1 + \exp(-y_i w^\top x_i)} > \frac{1}{k}$$

We consider the under-approximation of the RLO hessian coefficient given by ignoring the second term inside the square parenthesis in equation 15. This is the main novelty in our analysis. The under-approximated hessian can be written as,

$$\underline{\nabla}^2 \mathcal{L}_{\text{RLO}}(w; x_i, y_i) = \frac{1}{k} \cdot \left[ \frac{\exp(-y_i w^\top x_i)}{\left(1 + \exp\left(-y_i w^\top x_i\right)\right)^{1 - \frac{1}{k}}} \right] x_i x_i^\top. \tag{16}$$

Let us use $r_i$ to denote the ratio of under-approximation of hessian coefficient in equation 16 of RLO and ratio of hessian coefficient of LR in equation equation 15. Recall that we need $r_i > 1$ for some $k$ to finish the proof. Now we proceed with the calculation of $r_i$ as follows,

$$r_i := \left[ \frac{1}{k} \cdot \frac{\exp(-y_i w^\top x_i)}{\left(1 + \exp\left(-y_i w^\top x_i\right)\right)^{1 - \frac{1}{k}}} \right] \div \left[ \frac{\exp(-y_i w^\top x_i)}{\left(1 + \exp(-y_i w^\top x_i)\right)^2} \right]$$

$$= \frac{1}{k} \times \frac{\cancel{\exp(-y_i w^\top x_i)}}{\left(1 + \exp\left(-y_i w^\top x_i\right)\right)^{1 - \frac{1}{k}}} \times \frac{\left(1 + \exp(-y_i w^\top x_i)\right)^2}{\cancel{\exp(-y_i w^\top x_i)}}$$

$$= \frac{\left(1 + \exp(-y_i w^\top x_i)\right)^{1 + \frac{1}{k}}}{k}. \tag{17}$$

Now, note that $r_i > 1$ in the above equation equation 17 if and only if,

$$r_i > 1 \iff \left(1 + \exp(-y_i w^\top x_i)\right)^{1 + \frac{1}{k}} > k \iff \left(1 + \frac{1}{k}\right) \log\left(1 + \exp(-y_i w^\top x_i)\right) > \log(k)$$

Note that the last inequality in the above equation is satisfied if

$$\left(1 + \frac{1}{k}\right) \cdot \log\left(1 + \exp(-y_i w^\top x_i)\right) > \log\left(1 + \exp(-y_i w^\top x_i)\right) > \log(k). \tag{18}$$

Using the last inequality, we see that $r_i > 1$ if $k \leq \left(1 + \exp(-y_i w^\top x_i)\right)$, and we have the desired result. This concludes the proof of Lemma 2 in the main paper. □

Furthermore, by summing over the dataset or indices $i$, and minimizing over $w$ we can get an upper bound on $k$ in terms of the total loss function also. With this we can say that RLO is strictly better than LR objective from the optimization perspective.

## A.2 DATASET STATISTICS

We use the following datasets listed in Table 4 and 5 in our experiments.

## A.3 MORE IMPLEMENTATION DETAILS

We provide the example codes of our proposed RLO in Figure 6. Figure 6a is to calculated rooted loss and gradients following Eq 4 and 7. Figure 6b is to obtain rooted loss and can be optimized by any optimizer in PyTorch, which is easy to use for training deep neural networks.

## A.4 MORE EXPERIMENTS RESULTS

In this section, we include more experimental results and analysis of our proposed RLO loss function in multiple settings and applications.

### A.4.1 ROOTED LOGISTIC REGRESSION

We show the empirical results of rooted logistic regression on synthetic spiral dataset. Figure 7 shows that rooted logistic regression converge quickly in all settings. RLO with $\ell_2$ regularization also outperforms standard logistical regression in term of accuracy on test set. Specifically, RLO with $\ell_2$ regularization achieves 73.2% and LR with $\ell_2$ regularization achieves 73%. Hence, we conclude that our proposed rooted logistic regression is beneficial to accelerate the training and also provide improvements even in the simple synthetic setting.

```python
def root_grad(x):
    temp = 1. / (1. + np.exp(b * np.dot(A, x)))
    templ = np.power(1. + np.exp(-b * np.dot(A, x)), 1/self.kparam)
    grad = - np.dot(A.T, b * temp * templ) / n + lbda * x
    return grad

def root_loss(x):
    bAx = b * np.dot(A, x)
    return self.kparam * (np.mean(np.power(1. + np.exp(- bAx),1/kparam))) + lbda * norm(x) ** 2 / 2.
```

(a) Example code to calculate rooted loss and gradients.

```python
def root_loss(output, target, k, m):

    n = target.shape[0]
    prob = F.softmax(output, dim=1)
    root = torch.pow((prob[range(n), target]), 1 / k)
    root = m * (1 - root)
    loss = torch.mean(root)

    return loss
```

(b) Example code for using PyTorch optimizer.

Figure 6: The example code block to use our proposed RLO.

| Dataset | #Samples | #Attributes | #Classes |
|---------|----------|-------------|----------|
| Wine | 178 | 13 | 3 |
| Specheart | 267 | 44 | 2 |
| Ionosphere | 351 | 34 | 2 |
| Madelon | 4400 | 500 | 2 |

Table 4: Dataset information for regression in Section 4.2.

| Dataset | #Images | #Image size | #Classes |
|---------|---------|-------------|----------|
| CIFAR-10 | 60,000 | 32 | 10 |
| CIFAR-100 | 60,000 | 32 | 100 |
| Tine_ImageNet | 100,000 | 64 | 200 |
| Food-101 | 101,000 | 512 | 101 |
| Stanford Dogs | 20,580 | - | 120 |
| FFHQ | 70,000 | 1024 | - |

Table 5: Dataset information for image classification in Section 4.3 and GAN in Section 4.4.

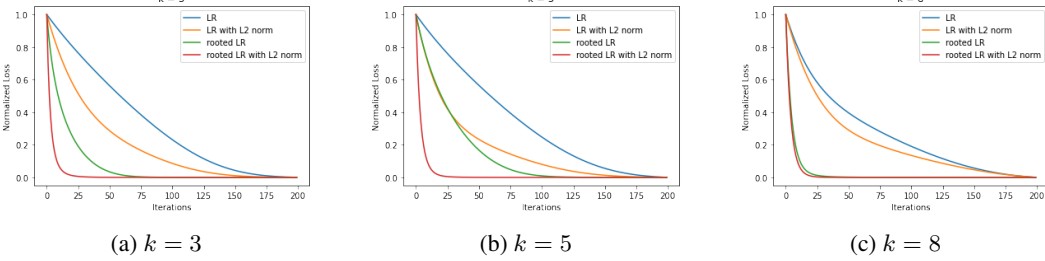

(a) $k = 3$      (b) $k = 5$      (c) $k = 8$

Figure 7: The rate of convergence over iterations of standard logistic regression and rooted logistic regression with different k on synthetic dataset.

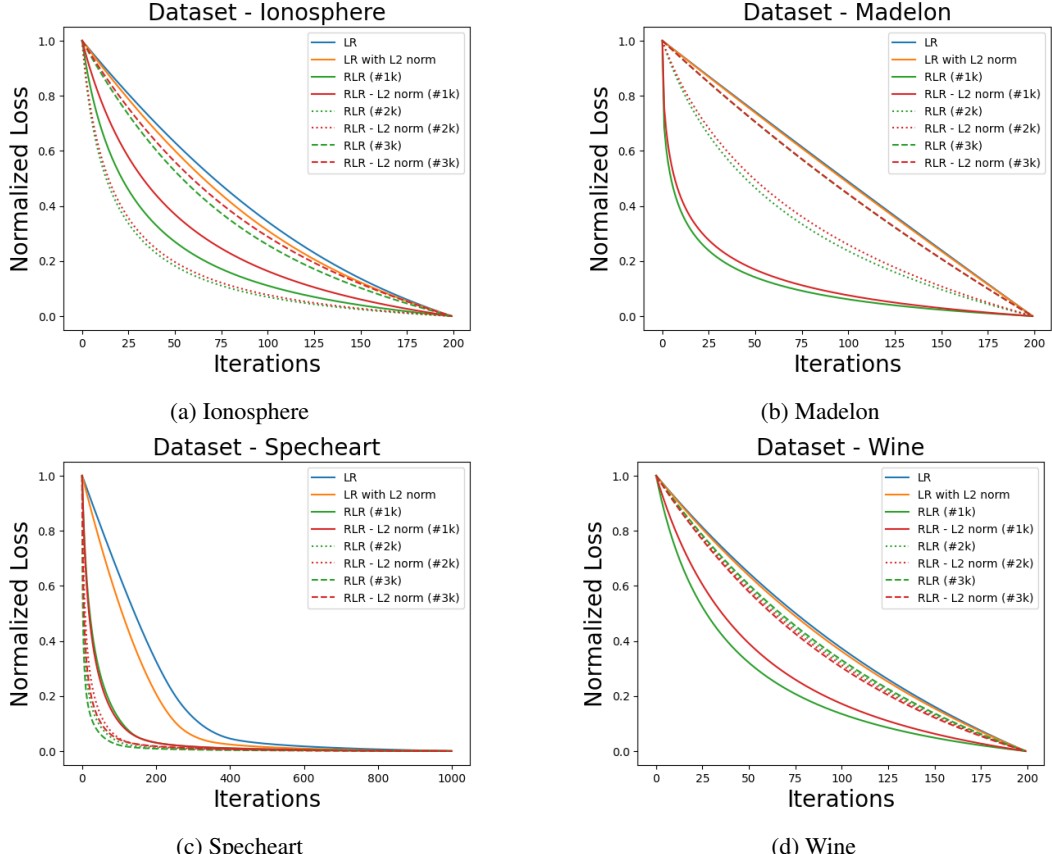

Figure 8: The rate of convergence over iterations of standard logistic regression and rooted logistic regression, for the train sets of Ionoshphere, Madelon, Specheart and Wine. The lines for the rooted logistic regression show the convergence for the value of top 3 $k's$ which gives the best test accuracy, as mentioned in Table 1. For RLO, solid lines show the normalized loss with the best $k$ value. The dotted and dashed lines of the same color depict the second and the third performing $k$ value.

### A.4.2 SHALLOW LOGISTIC REGRESSION VS ROOTED LOGISTIC REGRESSION

**Detailed experiments setups:** The Wine dataset contains 3 classes, so we use the One-vs-All approach for binary classification. The reported results are obtained by averaging the results from these three separate binary classification tasks. All the other datasets contains 2 classes, and undergo the standard binary classification task using the 4 regression methods. The classification performance for all the different settings is performed using K-fold cross-validation, with the number of folds being 5. The results shown are averaged across the 5 folds.

**More analysis:** Figure 8 shows the convergence performance for all the top 3 performing k values of RLO, with/without $\ell_2$ regularization, compared to LR with and without $\ell_2$. Again, it clear that RLO helps in faster convergence, compared to the standard logistic regression for Wine, Ionosphere, Madelon and Specheart datasets.

### A.4.3 MORE RESULTS OF IMAGE CLASSIFICATION WITH RLO

**Detailed experiments setups:** In the synthetic settings for binary classification, we implemented three different layers (2, 3, 4) fully-connected neural networks (FCN). The training iterations are 1000, 100, and 50 respectively. We use the same hidden size of 100, learning rate as 0.01 and k of 3 for three FCNs. For the vision models in image classification tasks, as multi-class classification, we train and finetune on ViT-B (Dosovitskiy et al. (2020)), ResNet-50 (He et al. (2016)), and Swin-B (Liu et al. (2021)) models. The k parameters of our proposed RLO are chosen from the set {5, 8, 10}.

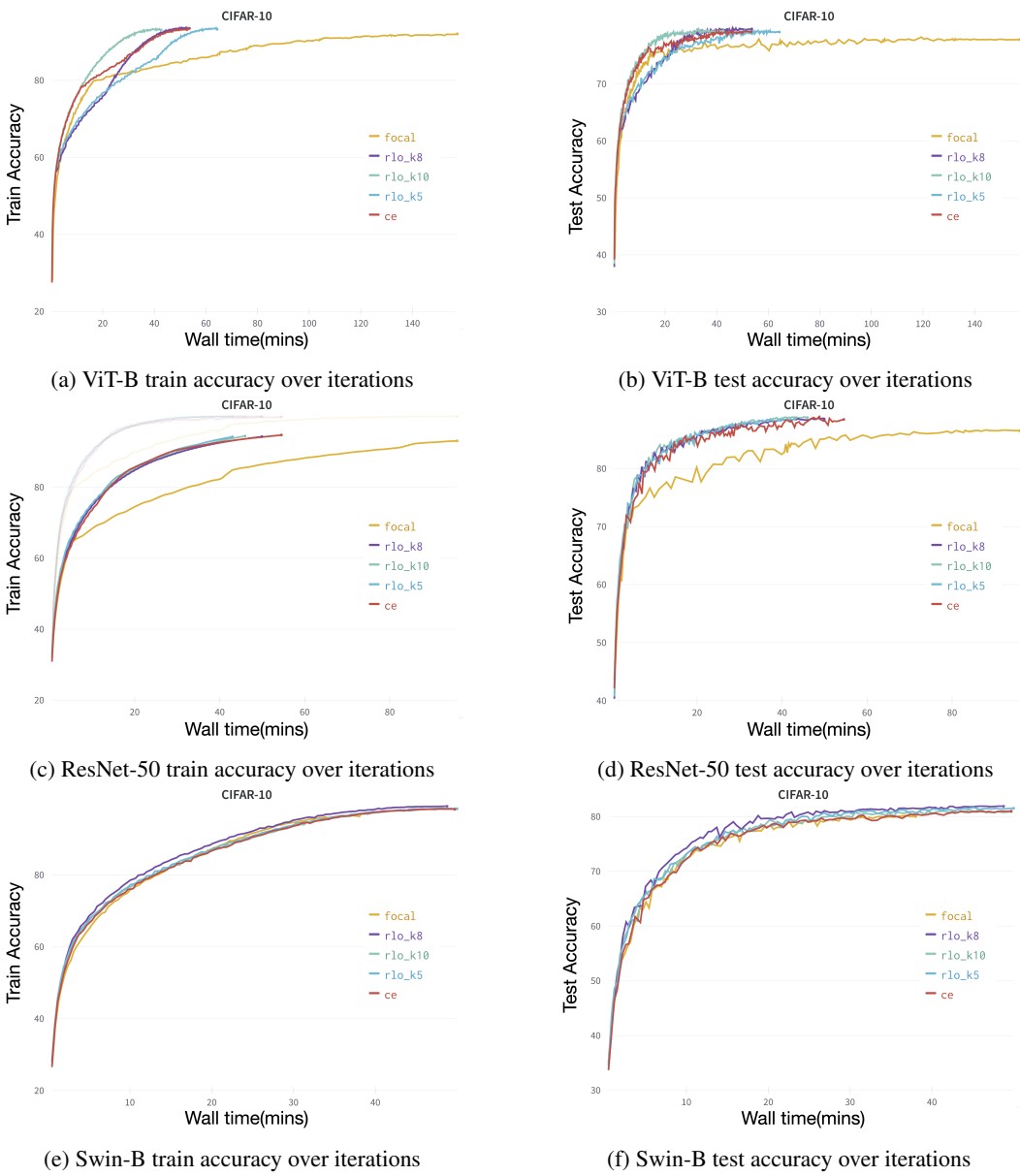

(a) ViT-B train accuracy over iterations

(b) ViT-B test accuracy over iterations

(c) ResNet-50 train accuracy over iterations

(d) ResNet-50 test accuracy over iterations

(e) Swin-B train accuracy over iterations

(f) Swin-B test accuracy over iterations

Figure 9: Train and test accuracy over iterations of different models on CIFAR-10. $k$ values are 5, 8, and 10. RLO outperforms on all models on both train set and test set.

We train the three models on CIFAR-10 and CIFAR-100 for 200 epochs with ViT and 100 epochs with ResNet and Swin. The batch size is 512 and learning rate is 1e-4. Moreover, we finetune the models which pre-trained on ImageNet (Deng et al. (2009)) on Tine-ImageNet and Food-101 for 10 epochs with batch size of 256 and learning rate of 1e-5. We use AdamW optimizer designed by (Loshchilov & Hutter (2018)). In addition, we apply RandAugment (Cubuk et al. (2020)) as augmentation strategy in finetuning steps. We train and fine-tune both on 3 NVIDIA RTX 2080Ti GPUs. To evaluate the effectiveness of our proposed RLO, we use cross-entropy (CE) loss and focal loss as baselines.

**More analysis:** Figure 9 shows train and test accuracy over iterations for ViT-B, ResNet-50 and Swin-B on CIFAR-10. RLO significantly reduces the training time and also provides performance improvements in term of train and test accuracy on all models.

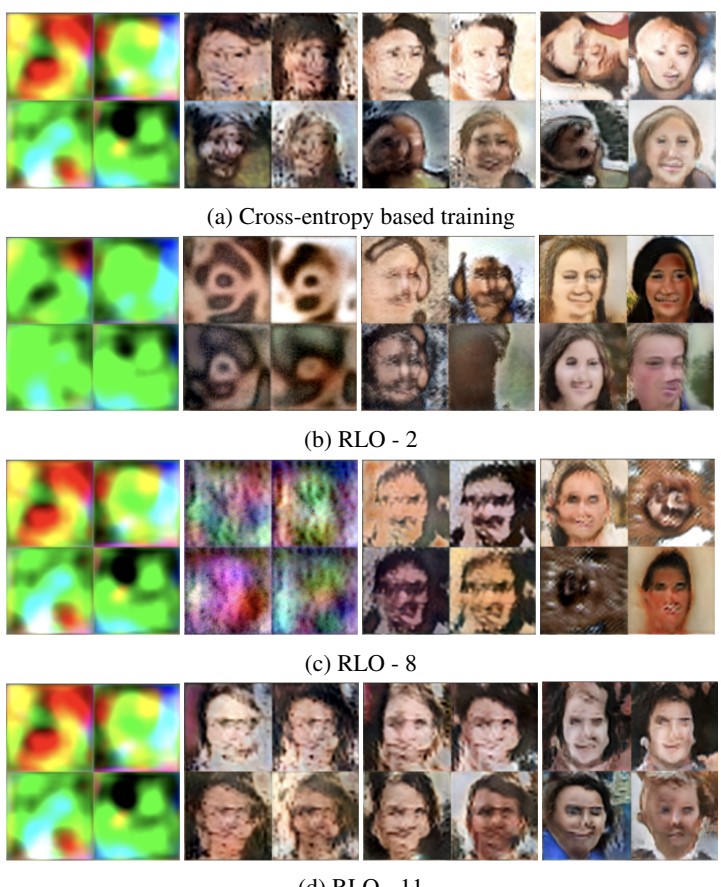

(a) Cross-entropy based training

(b) RLO - 2

(c) RLO - 8

(d) RLO - 11

Figure 10: The progressive generation of images shown with cross-entropy loss and RLO at different values of $k$ using the FFHQ dataset.

### A.4.4 MORE RESULTS OF IMAGE GENERATION WITH RLO

**Detailed experiments setups:** For the image generation setup, we use StyleGAN which is capable of being trained by limited training data. All training is run on 3 NVIDIA RTX 2080Ti GPUs, using 200 $64 \times 64$ dimension images from the FFHQ dataset, and 55 $64 \times 64$ images from the Stanford Dogs dataset. We use a mini-batch of 32 images and learning rate of 1e-4, for both the baseline, as well as our setup. We further evaluate the efficacy of RLO by replacing the original loss with our derived rooted loss function and compare it to StyleGAN's cross-entropy loss, for different values of $k$. To compare the efficacy of the models trained using RLO and cross-entropy loss, we take a large image from the original dataset, and compute its projection on the latent space using our model snapshots from the initial and final stages of the training. We then use these projections to generate an image using their respective models.

**More analysis:** We demonstrate more progressively generated images in Figure 10 for FFHQ dataset. We show the generated images with different values of $k$ and also compare to the cross-entropy which is our baseline. The results show that our proposed RLO is able to generate high-quality images with training on limited data.