# OpenReview forum: "Accelerated Neural Network Training with Rooted Logistic Objectives"
_ICLR.cc/2024/Conference — ICLR 2024 Conference Withdrawn Submission_

### Official Review · Reviewer_G5jt · 2023-10-30

**Soundness:** 3 good
**Presentation:** 2 fair
**Contribution:** 2 fair
**Rating:** 3
**Confidence:** 4

**Summary:**

This paper proposes the rooted logistic loss function for supervised classification and unsupervised generation tasks with nice theoretical properties. Moreover, this paper shows the comparison results of their rooted loss, cross-entropy loss and focal loss being applied to various datasets.

**Strengths:**

1. This paper proposes a new rooted loss objective.
2. This paper is sound.
3. This paper provides a lot of experiments on different datasets of supervised and unsupervised learning.

**Weaknesses:**

1. The contribution of this paper is limited.
The new objective loss function is based on the approximation of the natural logarithm function. If using the proposed loss objective, we introduce a new tuning parameter $k$. It may take a lot of time to tune this new parameter, but the improvement of the performance is not significant enough, as shown in Table 3.
2. Some parts of the paper are not explained clearly.
For example, this paper mentions that the reason of generalization bounds for logistic regression still holding for rooted logistic objectives is when $k\rightarrow +\infty\cdots$. However, from Lemma 2, a small $k$ is needed for fast convergence. Is there any contradiction here?
3. The presentation of the paper can be improved. For example, in Table 1, the results can be presented with $k$ ordered for each dataset (from small to large or reversely). There are typos in formulas (5), (6) and (7).

**Questions:**

1. In Table 1, I observe that for RLO, we can get better test accuracy with smaller $k$. But for ROL-L2, we get better test accuracy with larger $k$. Is there any insights about the reason for that? And how did the authors choose $k$ for each dataset?

---

> ### Author Response · Authors · 2023-11-22
>
> Dear Reviewer G5jt,
>
> Thank you for dedicating your time and expertise to review our paper on the rooted logistic loss function. We sincerely appreciate your thorough analysis and constructive feedback.
>
> Your acknowledgment of the novelty of our proposed loss function and the comprehensive experimental setup we employed is greatly valued. These aspects are central to our work, and we are glad to see them recognized.
>
> Your questions about the test accuracy in relation to the tuning of the new parameter are insightful and will guide us in providing more detailed explanations and insights in our future revisions.
>
> Your feedback is invaluable to us in refining our work, and we are grateful for the opportunity to improve our research based on your suggestions.
>
> Sincerely,
> Authors

---

### Official Review · Reviewer_MpCx · 2023-10-31

**Soundness:** 2 fair
**Presentation:** 2 fair
**Contribution:** 3 good
**Rating:** 6
**Confidence:** 3

**Summary:**

The paper proposes a novel loss function termed the Rooted Logistic Objective function (RLO) which builds upon the approximation of the logarithm function. Furthermore, the research establishes that under specific hyperparameter constraints, the proposed method exhibits strict convexity. The paper includes experiments from various domains and tasks to demonstrate the effectiveness of the proposed method.

**Strengths:**

1. The paper is overall well-organized.
2. The proposed method is interesting and novel to the best of the reviewer's knowledge.

**Weaknesses:**

1. The experiments conducted in the paper are based on datasets that are too simple and the corresponding baseline test accuracy is not reasonable (<90% test accuracy for Cifar-10). The reviewer would appreciate if the results of more realistic datasets could be included.
2. The reviewer didn't check the full derivation in the Appendix, but the derivation shown in the paper seems a bit sloppy (see Questions), hence hindering the soundness of the paper a bit.

**Questions:**

1. In Section 3.1, the authors wrote "Moreover, when we consider the gradient...", should the word "moreover" be rephrased to other words like "however" or "nevertheless"?
2. From (5) to (6), why is the term $\frac{1}{k}$ dropped?
3. In (6), should the right-hand side be $\ell_i^k(w) \cdot \frac{1}{exp(yw^Tx) + 1} \cdot (-yx)$?
4. In the discussion after Lemma 2, the authors wrote "It is beneficial when using stochastic algorithms that use a random mini-batch of samples at each iteration instead of the full dataset to compute gradient", why is this the case for RLO?
5. Why does RLO show faster training time compared with CE? The reviewer can understand RLO takes less iteration to convergence, but why is the training time shortened with a fixed number of iterations?
6. In Table 3, the test accuracy for CIFAR-10 on ResNets is noted to be below 90%. However, based on the reviewer's knowledge,  the test accuracy should be at least 93% with appropriate training setups.
7. On page 9, the authors introduce an additional parameter $m$ to be the multiplier for (4), is $m$ a scalar multiplied by the loss? Could the authors clarify more?

---

> ### Author Response · Authors · 2023-11-22
>
> Dear Reviewer MpCx,
>
> Thank you for your comprehensive and insightful review of our paper on the Rooted Logistic Objective function. Your detailed analysis and thought-provoking questions have provided us with valuable perspectives that will undoubtedly enhance the quality and clarity of our research.
>
> We appreciate your commendation on the organization of the paper and the novelty of our proposed method. Your feedback regarding the need for more complex datasets and a clearer presentation of our mathematical derivations is well-received. We agree that these aspects are crucial for the robustness and credibility of our work.
>
> We will address each of your questions in our revised manuscript to ensure that our methodology and findings are presented as clearly and accurately as possible. After our revision, we hope the support and enthusiasm will be much greater than the current level.
>
> Sincerely,
> Authors

---

### Official Review · Reviewer_vuQU · 2023-10-31

**Soundness:** 2 fair
**Presentation:** 2 fair
**Contribution:** 1 poor
**Rating:** 3
**Confidence:** 4

**Summary:**

This paper provides a new loss function named the Rooted Logistic Objective function in the logistic regression problem. Numerical results show that minimizing RLO achieves faster convergence rate and improved generalization than traditional loss function, e.g., cross-entropy (CE).

**Strengths:**

The presentation of this paper is clear.

**Weaknesses:**

1. RLO lacks the mathematical derivation by replacing the log with $1/k$ in the cross-entropy loss. In fact, the cross-entropy is to minimize the -log P, where $P$ is the likelihood. As the data are i.i.d, the likelihood of all the data can be written as the multiplication of the likelihood of each sample, e.g., $$\log P(y_1,y_2,...,y_n|x_1,x_2,...,x_n)= \log \prod P(y_1|x_1)...P(y_n|x_n) = \sum_{i=1}^n P(y_i|x_i).$$
In this paper, it replaces log with $(\cdot)^{1/k}$ and still sums them together. What is the intuition behind this? I can understand it is to maximize the summation of the likelihood of each sample, but it does not make any sense.

2. This paper focuses on non-linear models, specifically neural networks. However, the theoretical analysis in Section 3.2 is based on linear models. When considering non-linear models, both RLO and CE become non-convex functions with respect to the parameter $w$. In non-convex optimization, our primary concern is determining which objective function is easier to find the global optimum.

3. I did not see significant improvement via using RLO from Figure 4 and table 3.

**Questions:**

Why is the test error for Cifar-10 in Table 3 so high (compared with SOTA methods)?

---

> ### Author Response · Authors · 2023-11-22
>
> Dear Reviewer vuQU,
>
> Thank you for your thoughtful and detailed review of our paper on the Rooted Logistic Objective function. We greatly appreciate the time and effort you took to analyze our work and provide constructive feedback.
>
> Your comments on the need for a clearer mathematical derivation and the relevance of our approach to non-linear models will be addressed. These insights will guide us in refining our research and addressing the areas that require further exploration and explanation.
>
> We also acknowledge your concerns regarding the empirical evidence presented and are motivated to conduct more rigorous testing to validate our claims. Your feedback has been instrumental in highlighting the aspects of our study that need improvement, and for that, we are truly grateful.
>
> Sincerely,
> Authors

---

### Official Review · Reviewer_QRfD · 2023-11-02

**Soundness:** 2 fair
**Presentation:** 2 fair
**Contribution:** 2 fair
**Rating:** 3
**Confidence:** 4

**Summary:**

Aiming to better condition the logistic loss function, this work proposed Rooted Logistic Objective (RLO), a very simple polynomial approximation to the log function with an additional power parameter k to control the approximation.
Such a loss can serve as a plug-in replacement for log-based loss functions for supervised classification and unsupervised generation tasks.

Theoretically, the RLO objective is shown to be strictly convex whenever k > 1, and guaranteed to be as conditioned as the Logistic objective function in terms of the Hessian term if k is not too large.

Empirically, the authors conducted a series of experiments on classification and StyleGAN training, involving both fully connected networks and transformers. The modified loss is shown to converge faster and can achieve better performance.

**Strengths:**

The RLO objective seems novel to me.
Advertised to be a better alternative to logistic loss, the proposed RLO can potentially be widely applied to various supervised tasks.
In this work, the authors not only evaluated standard image classification but also extended to training the discriminator in GAN models. I like the inclusion of a toy data case to provide more visual clues.

**Weaknesses:**

## Weak theoretical analysis

First, the scope of this work is on substituting the logistic loss, or more precisely, approximating the log function with polynomials. However, in classification, the logistic loss is only one of the many surrogate losses for the more fundamental 0-1 loss.  The authors did not discuss any other surrogate loss functions and how they relate to the 0-1 loss. [1] is a related work.

Second, the theoretical statements are hand-wavy. For instance, the "better conditioning" concept is not clear in this work.
The authors wrote that "From lemma 1 and 2, we can conclude that there is a range of values of k that provides better conditioning for individual data points." I don't think this statement is well supported. The relationship between the $h(,)$ function and how conditioned is the objective function is not clear. The reason why a larger $h$ indicates that "the gradient directions may provide sufficient descent needed for fast convergence" should be made more rigorous.

Also, Lemma 2 only concerns the optimal solution $w^*$, but not the optimization trajectory. So I think the theoretical results are relatively weak.



## Insufficient experiments.
The empirical evaluation in this work is the biggest concern for me.

First, the experiments are not consistent enough. For example, in Table 1, the effect of the L2 regularization is not consistent. Ideally, if the weight decay strength is appropriate, shouldn't it always be better than that without weight decay? To be more specific, in Ionoshphere, L2 improves acc for LR but hurts acc for RLO (k=3). These inconsistencies cast doubt on the validity of the results. I wonder whether the hyperparameters are well-tuned.

Second, for the real-data experiments, the reported results are far from state-of-the-art. Take the CIFAR-10 classification for example, the reported acc for ResNet is only 87.67, which is significantly lower than other reported results. The red curve for CE in Figure 3(b) seems obviously not well-tuned.
The same can be said in the StyleGAN experiments, the FIDs are so high that it may not make much sense.

I strongly recommend the authors to first reproduce some SOTA results for image classification and GANs, and then try to improve upon them to make a more convincing case.



## Misc.
Many typos. For example $L_{RLO}(w)$ after lemma 1 should be $L_{LO}(w)$.

"$x$ is a smooth function, $f(z) \approx x$ are not appropriate. $x$ is a random variable that comes from a distribution.


**Overall**, I think this work can be significantly improved if either

* (1) Conduct a more thorough empirical evaluation and showcase that RLO can actually improve SOTA methods. [2] is an example.

* (2) Make a more significant theoretical contribution by providing improved convergence bounds for the proposed RLO. [3] is an example.


Reference:

[1] Bartlett, Peter L., Michael I. Jordan, and Jon D. McAuliffe. "Convexity, classification, and risk bounds." Journal of the American Statistical Association 101.473 (2006): 138-156.

[2] Hui, Like, and Mikhail Belkin. "Evaluation of neural architectures trained with square loss vs cross-entropy in classification tasks." ICLR 2021.

[3] Hu, Tianyang, et al. "Understanding square loss in training overparametrized neural network classifiers." Advances in Neural Information Processing Systems 35 (2022): 16495-16508.

**Questions:**

See weakness

**Details Of Ethics Concerns:**

No ethics concerns.

---

> ### Author Response · Authors · 2023-11-22
>
> Dear Reviewer QRfD,
>
> We wanted to express my gratitude for your insightful and detailed feedback on our work regarding the Rooted Logistic Objective. Your constructive criticism, highlighting both the strengths and weaknesses of our approach, has provided us with a clearer direction for future improvements. We particularly appreciate your suggestions on conducting more comprehensive empirical evaluations and enhancing our theoretical contributions. We have plans to further polish the arguments regarding condition number of RLO in our future revision.
>
> We are committed to addressing the areas of concern you've pointed out and hope to present a more robust and well-rounded study in our subsequent iterations. Thank you once again for your valuable input and guidance.
>
> Sincerely,
> Authors